# A Mini Review on Natural Safeners: Chemistry, Uses, Modes of Action, and Limitations

**DOI:** 10.3390/plants11243509

**Published:** 2022-12-14

**Authors:** Xile Deng

**Affiliations:** Key Laboratory for Biology and Control of Weeds, Hunan Agricultural Biotechnology Research Institute, Hunan Academy of Agricultural Sciences, No. 2 Yuanda Road, Changsha 410125, China; dengxile@hunaas.cn; Tel.: +86-10-62890876

**Keywords:** herbicide, natural product, safener, mode of action, crop plant, metabolization, herbicide damage

## Abstract

Herbicide injury is a common problem during the application of herbicides in practice. However, applying herbicide safeners can avoid herbicide damage. Safeners selectively protect crops against herbicide injury without affecting the biological activity of herbicides against the target weeds. However, after long-term application, commercial safeners were found to pose risks to the agricultural ecological environment. Natural safeners are endogenous compounds from animals, plants, and microbes, with unique structures and are relatively environment-friendly, and thus can address the potential risks of commercial safeners. This paper summarizes the current progress of the discovery methods, structures, uses, and modes of action of natural safeners. This study also concludes the limitations of natural safeners and prospects the future research directions, offering guidance for the practical application of natural safeners to prevent herbicide injury. This study will also guide the research and development of corresponding products.

## 1. Introduction

Herbicides can efficiently prevent and control weeds, safeguarding the production of food crops and increasing crop yields. Besides reducing human and material costs, herbicides also bring enormous economic benefits to society [1,2,3,4]. However, herbicides cause herbicide injury. Improper use of herbicides can significantly affect agricultural production and even cause total crop failure. Herbicide safeners can effectively solve the problem of herbicide injury [5,6,7]. About 30% of herbicide preparations on the global market contain safener ingredients [8]. Herbicide safeners selectively protect crops against herbicide injuries without affecting the herbicidal activities of herbicides against the target weeds [9], and thus they enhance the safety and expand the application scope of herbicides [10]. Since the first commercial safener, 1,8-naphthalic anhydride (NA), was discovered by Hoffman to protect maize from the injury caused by herbicide thiocarbamate in 1971, there are over twenty commercial safeners were developed, such as dichlormid, fenclorim, fenchlorazole-ethyl, and flurazole, and some of their structures are depicted in Figure 1 [10]. However, commercial safeners may be “unsafe” mainly because they are usually used as “inert active ingredients” in the formulation of herbicide preparations and do not need to go through the pre-market registration process or the toxicology test for pesticide registration in most countries. As a result, their environmental behaviors and potential environmental risks may be unknown [8]. However, studies have shown that some commercial safeners pose potential environmental threats. For instance, benoxacor is highly toxic to aquatic organisms, such as aquatic autotrophic freshwater algae and zebrafish (*Danio rerio*) embryos, with LC_50_ values of 0.63 mg/L and 0.60 mg/L at 96 h, respectively. Dichloroacetamide safeners also produce low to medium toxicity in rats [11,12,13,14]. Therefore, it is necessary to identify novel eco-friendly safeners.

For more than 30 years, many research groups have been involved in studies to develop alternative strategies to combat weeds and parasitic plants that are very dangerous to important crops. These alternatives are based on the use of natural compounds and have the objective of reducing the environmental pollution generated by the massive use of chemicals in agriculture and the risks to human and animal health. The origin of these phytotoxins is essentially from fungi pathogens for weeds and from allelopathic plants. In some cases, suitable formulations of pure phytotoxins and/or crude extracts of fungi and plants were also developed, and some studies on their efficacy, selectivity, and toxicology were also carried out [15,16]. Natural herbicide safeners, with broad research and application values, are endogenous compounds from animals, plants, and microbes that have an advance in environmental friendliness. To date, various natural products can be used as safeners. This paper provides a systematic review of research progress on the discovery methods, structures, action mechanisms, and application potentials of different natural safeners. Therefore, it will provide references for discovering natural safeners, clarifying action mechanisms, and the research and development (R&D) of related safener products.

## 2. Chemistry

Table 1 shows the current natural safeners for protecting different crops. Gibberellin A_3_ (GA_3_) was the first reported discovered natural safener in 1990, and most of the natural safeners were reported in the 2010s. Similar to commercial safeners, they are combined used with safeners by seed treatment and pre- and post-emergence treatment. The reported natural safeners isolated from plants (for example, the Chinese medicinal herbs), fungus, and animals, include sanshools (San), echinacea alkylamides (ECHAAs), *Z*-ligustilide (*Z*-LIG), senkyunolide A (SNA), bergapten (BP), isopimpinellin (ISO), salicylic acid (SA), GA_3_, brassinolide (BR), methyl jasmonate (MeJA), methyl dihydrojasmonate (MDJ), melatonin (MEL), and dimboa.

As shown in Figure 2, we further evaluated the chemical structures of the reported natural safeners and found that they are different in chemical skeletons compared to those commercial ones (Figure 1). For those natural safeners extracted from Chinese medicinal herbs, their structures could be divided into three categories: (i) *N*-alkyl unsaturated amides (San and ECHAAs); (ii) Benzofurans (*Z*-LIG and SNA); and (iii) Furocoumarin compounds (BP and ISO). Interestingly, before the above compounds were identified as natural safeners, the current study indicated that they also have a wide range of medicinal and pesticidal activity, such as promoting blood circulation, antitussive activity, antitumor activity, insecticidal activity, and bactericidal activity [17,18,19]. For those natural safeners screened in the know natural chemicals, most of them are key hormones and signal molecules in plants that regulate many physiological and biochemical processes of plant growth and development and act as plant-derived growth regulators [20,21,22]. Almost all natural safeners could be obtained via biosynthesis and chemical synthesis, except for extracting from Chinese medicinal herbs [23,24,25,26,27,28,29,30,31]. 

**Table 1 plants-11-03509-t001:** Names, applied crop plants, application modes, first discovery time of natural safeners.

Safener	Crop	Application Mode	First Discovery Time	References
gibberellin	Maze, rice, proso millet, and sorghum	Seed treatment, pre- and post-emergence	1990	[32,33,34,35]
dimboa	Maize	Pre-emergence	1995	[36]
brassinolide	Maize, rice, and proso millet	Seed treatment, pre- and post-emergence	1998	[37,38,39,40,41]
salicylic acid	Maize, rice, and soybean	Seed treatment, Pre- and post-emergence	2004	[42,43]
melatonin	Rice, pea, and sweet potato	Post-emergence	2013	[19,44,45,46]
*Z*-ligustride	Rice	Pre-emergence	2013	[47,48]
senkyunolide A	Rice	Pre-emergence	2013	[47,48]
sanshools	Rice	Pre-emergence	2014	[49]
bergapten	Rice	Pre-emergence	2017	[50]
isopimpinellin	Rice	Pre-emergence	2017	[50]
echinacea alkylamides	Rice	Pre-emergence	2017	[51]
methyl dihydrojasmonate	Rice	Pre-emergence	2019	[52]
methyl jasmonate	Rice	Pre-emergence	2020	[53]

## 3. Uses

### 3.1. Natural Safeners for Chloroacetanilide Herbicide

Chloroacetanilide herbicides, such as alachlor (AI), acetochlor (AC), butachlor (BC), metolachlor (MET), and *S*-metolachlor (*S*-MET), which are one of the most popular herbicides in the world market (ranked No. 3), can cause significant phototoxicity and reduce yields to crop plants such as rice (*Oryza sative*), maize (*Zea mays*), proso millet (*Panicum miliaceum*), and sorghum (*Sorghum bicolor*) [10]. The most reported natural safeners are those for chloroacetanilide herbicides. In 1988, Wilkinson found that the application of exogenous GA_3_ to soybean (*Glycine max*) seedlings reversed the adverse effects induced by metolachlor (MET) [37]. It is the first recorded case of chloroacetanilide herbicide natural safener. GA_3_ also showed protection efficacy on maize against MET injury, indicating one natural safener can relieve the herbicide injury caused by the same herbicide to different crops [32]. With the exception of GA_3_, there are also several reported natural safeners (SA, San, and ECHAAs) for MET [43,49,51]. Although they are all safeners used for rice, it is difficult to compare the safening activities between them because the experimental concentrations of herbicides and safeners set in the references were not the same. However, San showed great potency against metolachlor damage at 0.8 mg/L. The shoot height, root length, and fresh biomass of rice were recovered to 93.1%, 97.6%, and 94.8% by San, compared to the rice only treated with MET, respectively [49]. The enantiomeric monomer of MET (*S*-MET), with higher herbicidal efficacy than that of MET, GA_3_, *Z*-LIG, SNA, MeJA, and MDJ, was reported as the natural safeners for it. For *Z*-LIG and SNA, which is extracted from *Ligusticum chuanxiong* as the main active ingredients, safener activity tests based on the agar medium method and soil medium method showed that both the two compounds were more protective of shoots than roots of rice seedling and had a significant difference in safener activities. It showed that *Z*-LIG had significantly higher safener activity than SNA, suggesting that Z-LIG may be the major active component in *Ligusticum chuanxiong* [48]. For MeJA and MDJ, MDJ showed greater safener activity against injury caused by *S*-MET in rice than MeJA [52,53]. The results above revealed that subtle differences in the chemical skeleton could cause significant differences in safener activity. Additionally, Choi et al. reported BR could enhance the resistance of rice seedlings to reduce the herbicidal phytotoxicity caused by butachlor via the seed soaking treatment [38]. Hu et al. found that BP and ISO can increase the AC tolerances of rice seedling shoots, and BP demonstrated a better protection effect than ISO. It should be mentioned that BP did not affect the weed control effect of AC on the target weed barnyard grass (*Echinochloa crusgalli*) [50]. This is the only research evaluating whether nature safeners can influence herbicidal activity. Notably, MeJA could also be the safener for AC [53]. 

### 3.2. Natural Safeners for Sulfonylurea Herbicides

Similar to chloroacetanilide herbicides, sulfonylurea herbicides also occupy an important position in the global herbicide market (ranked No. 2) [54]. Ethametsulfuron–methyl (EM) is a type of sulfonylurea herbicide for controlling post-emergence weed in oilseed rape *(Brassica napus*). The great damage induced by EM to crop plants such as rice and maize limited its further application. The experimental results under laboratory or field conditions indicated that applying BR can protect both rice and maize from the phototoxicity induced by ethametsulfuron–methyl via the seed-soaking treatment strategy. BR can also eliminate the phytotoxicity of sulfonylurea herbicide formulation Atlantis (mesosulfuron–methyl + iodosulfuron–methyl–sodium 3.6% water dispersible granule) at low dosage (450 L/hm^2^), and 11.5% reduction in phytotoxicity index to proso millet at high dosage (600 L/hm^2^) [39]. Additionally, when combined used with monosulfuron, it also reduced 20.27% injury, compared with the control treatment (only treated with monosulfuron) [35]. However, the safener activity of SA was not exhibited in the patent implicated by Bayer [42]; its analogs 4-OH SA and 4-F SA could be used as crop safeners without affecting the efficacy of herbicides by the seed and post-emergency treatments. These two SA analogs can be combined with sulfonylurea herbicide foramsulfuron to alleviate the herbicide injury to maize and soybean. They can reduce the herbicide injury to crop maize by up to 82% compared with the control check (non-treatment) [42]. 

### 3.3. Natural Safeners for Other Types of Herbicides

Recent studies lend color to this view that one natural safener can effectively alleviate the herbicide injury caused by different herbicides to the same crop. For example, except as the safener for chloroacetanilide herbicides treated with rice, MeJA can also reduce the phototoxicity caused by arylpicolinate herbicide halauxifen–methyl (HM) to rice [53]. 4-OH SA and 4-F SA can also be combined used with organic heterocyclic herbicide isoxaflutole to reduce its herbicide damage [42]. GA eliminated the phytotoxicity of low-dose herbicide formulation Quelex (alauxifen–methyl + florasulam 20% wettable powder) and a 40.9% reduction in phytotoxicity of high-dose Quele. However, the weed control effects decreased by 1.7–18.0% in the treatments of herbicides and safener combinations [39]. As a potential safener, low dose (0.1 mM) MEL can reduce the herbicide injury caused by bentazone by 30% and significantly increase sweet potato yield compared with the blank control (twice treatments) [45]. However, one natural safener might have many differences in protecting the herbicide damage from different herbicides to the same crop. For instance, rice seedlings treated with the combination of 0.25 mg/L MeJA and 0.073 mg/L *S*-MET showed better protection effects on shoot height and root length than that of 0.25 mg/L MeJA and 0.073 mg/L AC and 0.25 mg/L AC and 0.073 mg/L HM [53]. Dimboa showed no safening activity against AC injury, but diboa exhibited detectable protection against dithiocarbamate herbicide eradicane injury [36]. Studies on the detoxification effects of the same natural safeners on different herbicides in the same crop are still lacking, and more related research should be carried out.

## 4. Modes of Action

### 4.1. Induction of GSTs

In herbicide metabolism, herbicide molecules first undergo hydrolysis or oxidation involved with peroxidase (POD) and cytochrome P450 (P450s) to form new functional groups. As seen in Figure 3, the as-obtained metabolite then binds to glutathione (GSH) to form conjugates and be metabolized into a non-toxic substance. This procedure was catalyzed by glutathione-*S*-transferases (GSTs). Natural safeners can induce the expressions of GSTs genes to enhance the catalytic activities of GSTs. For instance, San can also activate GSTs-encoding genes (*OsGSTF12*, *OsGSTU3*, *OsGSTU39*, and *OsGSTU39*) in rice, thereby enhancing the activity of GSTs and alleviating MET injury [41]. Furocoumarines natural safeners, BP and ISO, could promote the metabolism of AC in rice by significantly enhancing the GSTs activities, alleviating the herbicide damage caused by AC to rice seedlings [42]. Research has demonstrated that SNA can enhance the metabolic detoxification capacity of rice against *S*-MET by inducing the expression of GSTs-related gene *OsGSTU2* in the roots, stems, and leaves, and improving the GSTs activity of rice seedlings treated with *S*-MET, thus alleviating the herbicide injury [43]. BR can enhance the GSTs activities in panicum miliaceum, promoting the formation of non-toxic conjugates of herbicide formulation benzoxazole ٠ dimethyl and GSH [35]. Although the action mechanisms of SA and its analogs acting as safeners are still unknown, studies have reported that SA can enhance GSTs activity in rice seedlings, which may serve as the potential action mechanism [55]. Nemours researchers indicated that enhancing the GSTs activities in crop plants to metabolize herbicides is the most well-known mechanism of natural safeners [56].

### 4.2. Reduction in the Oxidative Damage

Herbicides can also cause significant oxidative damage while inhibiting the target activities in crop plants [57,58]. San and MET co-treatment could recover the chlorophyll content of rice leaves, compared with the blank control only treated with water [41]. Further research found that San could effectively recover the root activity of rice seedlings treated with MET (the activities of antioxidant enzymes, such as peroxidase (POD), superoxide dismutase (SOD), and catalase (CAT) in the roots). Lin et al. found that GA_3_ can increase SOD activity and chlorophyII content of proso millet leaves and significantly reduce malondialdehyde content, thereby alleviating the herbicide injury caused by Atlantis and Quelex (20% halauxifen–methyl + florasulam wettable powder) [39]. Lin et al. also found that BR can enhance the activity of antioxidant enzymes (POD, SOD, and CAT) in panicum miliaceum leaves to maintain the cellular redox balance [57]. Additionally, MEL can alleviate herbicide injury to crops by reducing the membrane damage and lipid oxidation of plant leaves, stimulating the enhancement of the activity of antioxidant enzymes (SOD, CAT, POD, and APX), and increasing the contents of antioxidant metabolites (ascorbate, GSH, and proline) exposed to herbicide sites [21]. These indicated that the reduction in oxidative damage in crops is also a crucial mechanism of natural safener.

### 4.3. Induction of Signaling Pathway

Safeners can induce defense and detoxification gene expressions in crop plants to protect them from herbicide injuries, rendering herbicides non-toxic to crops, which indicates safeners can activate detoxification signal pathways for herbicides. Zheng et al. analyzed the action mechanism by which GA_3_ protects sorghum from the herbicide injury caused by *S*-MET based on plant hormone metabonomics and found that *S*-MET causes herbicide injury by affecting the levels of abscisic acid, GAs, and other plant hormones in grain sorghum. They also found that exogenous application of GA_3_ can effectively restore the hormone levels and corresponding proportions of auxin and GA_3_ by up-regulating the expression of abscisic acid synthesis-related genes *ZEP* and *NCE* [33]. It should be mentioned that this is the only reported case about the signaling pathway of the detoxification effects of natural safeners. 

## 5. Limitations

Although verities of natural safeners were discovered, none of them were commercialized and further launched into the market; even Bayer applied for and was authorized the relevant patent [42]. It might be due to the reason that existing natural safeners have low activity compared to commercial safeners and thus are used in large doses. For example, at least 8 mg/L GA_3_ was used to recover the herbicide injury of 0.25 μmol/L (0.073 mg/L) *S*-MET-treated rice seedlings completely [34]. However, in the commercial herbicide formulation (Dual II 231 Magnum^®^), the ratio of the active ingredient of herbicide MET and safener benoxacor is 17.6:1 [11]. Additionally, some natural safeners are difficult to synthesize and at high extraction costs. For instance, the synthetic procedures of the monomers of San, such as α/β-sanshool, hydrogen-α/β-sanshool, and γ-sanshool are rather complicated [23,24,25,26,27,28,29,30,31]. The cost of supercritical chromatographic extraction is relatively high, and the extraction efficiency of San is low [49]. The industrial production of GAs relies on microbial engineering, leading to its high cost. One of the strategies to solve the problem of complicated structures, the availability, and the high cost of natural safeners is the modification of the chemical structure. Our group analyzed the chemical skeleton of San and designed and synthesized three *N*-alkyl substituted derivatives (*N*-isobutyl substituted analogs (2a–k), *N*-(2-hydroxy-2-methylpropyl) substituted analogs (3a–k), and *N*-(2-methylallyl) substituted analogs (4a–k)), via the carbon chain shortening method in scaffold hopping approach. Bioassay results showed that the 2k and 4k derivatives, with simple structures and easy to synthesize, have a high activity similar to that of the control commercial safener dichlormid, with low toxicity to zebrafish embryos. Moreover, the two derivatives can be used as lead compounds or active fragments in the R&D of novel efficient and low-biological risk safeners using relatively convenient synthetic processes with relatively high yields (50–67%), as shown in Figure 4 [5,59,60,61]. The other case of the synthesized safeners from the natural product is shown in Figure 5. Jablonkai et al. synthesized a series of analogous compounds of dimboa and its degradation product moba, finding that most of these compounds showed much better saving activities than that of dimboa and moba against the injuries of herbicides AC and 4-(dichloroacetyl)-1-oxa-4-azaspiro[4.5]decane [36].

## 6. Conclusions and Perspectives

Although safeners are crucial in preventing herbicide injury in practical production, existing commercial safeners are associated with many environmental risks after long-term use. However, the emergence of natural safeners offers a promising way to solve this problem. They have a unique chemical structure, and some are hormones or signal molecules that regulate key physiological and biochemical processes in higher plants during growth and development. These growth-related hormones can effectively counteract the negative effects of herbicides on crop development. Moreover, some natural safeners do not affect herbicide activity. One natural safener can effectively alleviate the herbicide injury caused by different herbicides to the same crop and herbicide injury caused by the same herbicide to different crops. However, existing natural safeners have low activity and thus are used in large doses. Natural safeners are also difficult to synthesize with high extraction costs, thus limiting their further field promotion, application, and commercialization. Therefore, it is necessary to restructure these commercial safeners. Some structurally simple natural safeners, such as SA, San, and MeJA, can be used as lead compounds or active fragments or modified via intermediate derivatization, active substructure splicing, and other pesticide design methods in combination with other highly active safener fragments to find novel, eco-friendly candidate herbicide safener compounds with a unique structure.

Moreover, in-depth studies assessing the action mechanisms of natural safeners are needed since most existing studies mainly focus on the effects of natural safeners on the metabolism of herbicides. Natural safeners can induce the enhancement of the activity of various key enzymes, such as GST and antioxidant enzymes (SOD, CAT, and POD) in the whole detoxification pathway in plants, thus boosting the metabolism, degradation, and separation of herbicides in crops. However, the molecular action mechanisms of safeners involve complex interactions between multiple signals and detoxification pathways. Additionally, other potential action mechanisms and pathways for natural safeners, such as the induction of ABC transporters, the effect on target enzyme activity, and the specific signaling pathways of induction, require further studies. The action mechanisms of natural safeners should be identified to improve the R&D of novel safeners with high efficiency, high safety, and strong selectivity based on such safeners. Future studies should evaluate the action mechanisms of natural safeners based on the following aspects: (1) transcriptomics, metabonomics, proteomics, and other advanced technical means emerging in recent years should be used to assess the effects of natural safeners on various key factors (enzymes, hormones, etc.) involved in the metabolism of herbicides; (2) the signaling pathways of natural safeners should be intensively studied to clarify how natural safeners regulate the expression of detoxification genes; (3) other potential action mechanisms of safeners, such as the induction of cell wall ABC transporters and the effect on the target enzyme activity of herbicides should also be analyzed.

## Figures and Tables

**Figure 1 plants-11-03509-f001:**
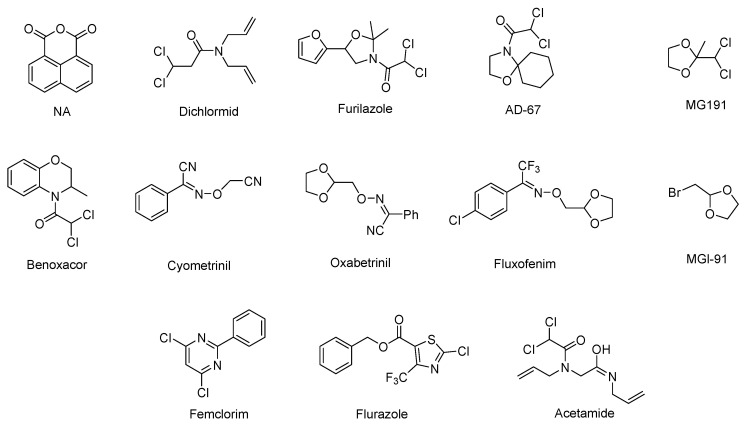
Chemical structures of some commercial herbicide safeners.

**Figure 2 plants-11-03509-f002:**
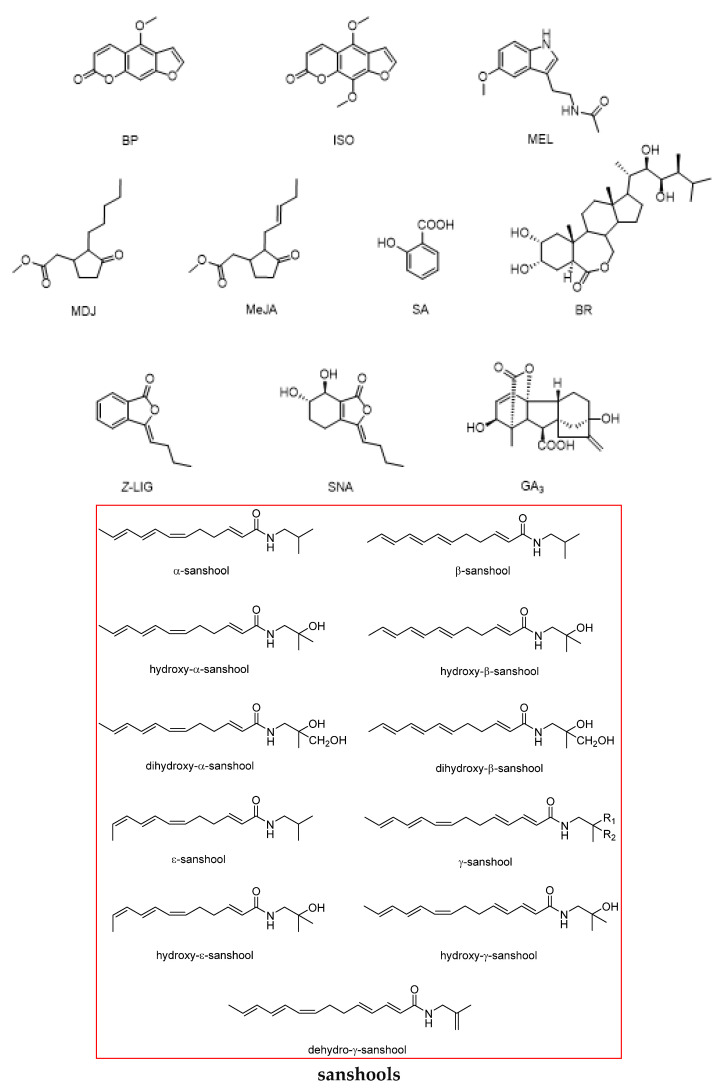
Chemical structures of reported natural safeners (sanshools, echinacea alkylamides, Z-ligustilide, senkyunolide A, bergapten, isopimpinellin, salicylic acid, gibberellin A_3_, brassinolide, methyl jasmonate, methyl dihydrojasmonate, melatonin, dimboa, and mboa).

**Figure 3 plants-11-03509-f003:**
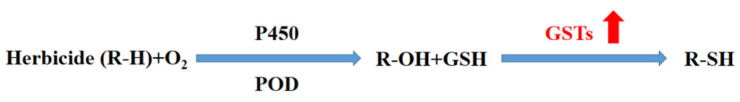
Induction of GSTs to detoxify herbicides (P450: cytochrome P450; POD: peroxidase; GSH: glutathione; GSTs: Glutathione *S*-transferase).

**Figure 4 plants-11-03509-f004:**
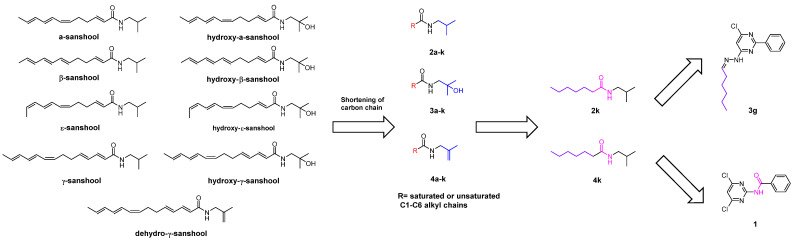
Strategies for structural modifications of natural safener sanshools and the as-obtained candidates with high activities (**2k**, **4k**; **3g**, and **1**).

**Figure 5 plants-11-03509-f005:**
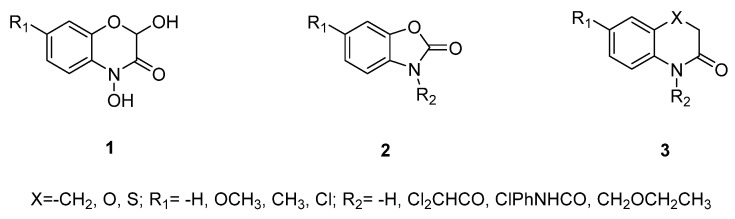
Analogues of dimboa (R_1_ = -H) and its degradation product moba (R_1_ = -OCH_3_, R_2_ = -H).

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
