# Peer review of "A Mini Review on Natural Safeners: Chemistry, Uses, Modes of Action, and Limitations"

_plants, 2022, doi:10.3390/plants11243509_

Round 1

Reviewer 1 Report

This study is aimed to review natural safeners including their chemistry, uses, mode of actions, and limitations. The study design is acceptable. The study contains some interesting points that can be worth to publish but some points need to be revised before considering for possible publication.

Comments/suggestions:

Delete points after section title titles. E.g. 4. Mode of action or 3.1. Natural safeners for chloroacetanilide herbicide.

L54: The title for figure 1 is too shotórt and not informative.

L88: Table 1: The title is too short and not informative. Application mode and not Application Mode.

L91: Figure 2: Give more information in the title.

L97: Again, the title is too short and not informative. Giev explanation for San and for ECHAAs

L195: Give explanations for all abbreviations e.g. GST, POD, GSH …

L250: Again, the title is too short and not informative. The letter size is too small for this figure.

Literature

L358: Zea mays – in italic

Reviewer 2 Report

The mini review titled “A mini review on natural safeners: chemistry, use , mode of actions, and limitation
reported briefly some information on the so called “safeners” that could be interesting for the farmer and
the scientists involved in the control of weeds.

Although the term”safener” is well defined also on the media among the scientists involved in the
biocontrol of weeds and parasitic plant is not common and extensively used. Thus the reivew is useful not
only for these scientists but also for the farmers thus I would give some suggestion to the authors:

1) since more than 30 years, many research groups are involved in the studies to developed alternative
strategies to combat weeds and parasitic plants very dangerous for important crop. These alternative are
based on the use of natural compounds and have the objective to reduce the environmental pollution
generated by the massive use of chemicals in agriculture and the risks for human and animal health. The
origin of these phytotoxins is essentially from fungi pathogen for weed and from allelopathic plants. In
some case also suitable formulation of pure phytotoxins and/or crude extract of fungi and plants were
developed as well as some study on their efficacy, selectivity and toxicology were also carried out. These
important aspects on the biocontrol of weeds should be cited in the manuscript reporting appropriate
literartures (see for example : Nat. Prod. Rep., 2015, 32, 162; Natural Product Communications Vol. 10 (6)
2015 1121);

2) why the author extensilvely treated only about the possibility to use extract of Chines medicinal plants to
be applied as safeners. It is very strange that they referred only to this herb that are known for applications
in medicine and do not include other suitable Chinese plants as week as that present in other region of the
world and containing in good amount safeners. There are 3525 references reported for safeners and many
was from different plants. This part should be integrate and appropriate literature cited.

3) From the data discussed it should be better clarified about the selective effect of safeners on weeds in
respect to that on agrarian plants. This is necessary as this selectivity seems very poor as well as the
availability of safeners and data on their toxicology;

4) The synthesis of some useful safeners, which was developed to overcome the low availability, is an
important practical aspect that should be more developed also reporting if the synthetic strategies have
high yield and are convenient and ecofiendly processes;

5) the paragraph reporting the mode of action seems only supported by a general hypothesized
mechanism. Thus this part should be clarified and supported with more appropriate literatures.

Round 2

Reviewer 2 Report

I tlhank the authors for the acceptance of some suggestion  but some questions remain to be satisfy

Point 1. the literatures 16 is  cited in the test but not in the end list

Point 2. this important point is not appropriately  integrate and revised

Point 4. The author limitate their revsion to only one synthesis. I think that that thes important point should me more ntegrated reprting the deatils of synthesis of other safeners
